# Acceptability and usability of COVID-19 antigen self-test in populations under socioeconomic vulnerability in Brazil: A cross-sectional study

Débora Castanheira[1], Laio Magno[2,3], Thais Aranha Rossi[2,4], Suelen Seixas[2], Fabiane Soares[4], Daniele Novaes[1], Ines Dourado[4], Valdilea G. Veloso[1], Thiago S. Torres[1]*

**1** Instituto Nacional de Infectologia Evandro Chagas, Fundação Oswaldo Cruz, Rio de Janeiro, Brazil, **2** Universidade do Estado da Bahia, Salvador, Bahia, Brazil, **3** Instituto Gonçalo Moniz, Fundação Oswaldo Cruz, Salvador, Bahia, Brazil, **4** Instituto de Saúde Coletiva, Universidade Federal da Bahia, Salvador, Bahia, Brazil

\* thiago.torres@ini.fiocruz.br

## Abstract

Rapid diagnostic self-tests have emerged as effective tools for identifying and controlling the spread of SARS-CoV-2. However, little is known about their acceptability and usability among populations under socioeconomic vulnerability globally. We aimed to evaluate the acceptability and usability of a COVID-19 antigen self-test among persons living in Salvador and Rio de Janeiro, Brazil. In this cross-sectional study (January–May 2023), participants used a COVID-19 antigen self-test in a simulated real-world setting, guided by an instructional video. Usability was assessed through two main outcomes: comprehension of instructions and proper execution of self-test (Poisson regression model); and accuracy in interpretation of self-test results (Cohen's kappa). Acceptability was evaluated based on willingness to reuse the self-test, user experience, and recommendation to others. Among 437 participants, most were women (65.7%), self-identified as Black/*Pardo (mixed-race)* (81.5%), aged 35+years (65.7%), had a household income ≤USD 470 (70.0%) and had completed secondary education (46.2%). Despite some procedural difficulties, most participants obtained valid results (88.1%), higher in Salvador (95.2%) than in Rio de Janeiro (81.7%) (p<0.001). Participants showed difficulty interpreting test results, particularly inconclusive with a positive mark (32.9% correct) and faint positive markers (25.2% correct). Accuracy in interpretation was 89.6%, with moderate to substantial inter-rater agreement (Cohen's kappa=0.56 overall, reaching 0.78 among participants aged 35–44 years). Participants with older age, lower education level and self-identified as men had lower likelihood of obtaining valid results in adjusted model, and increased difficulties in test setup and interpretation. Over 95% of participants were willing to reuse and recommended the self-test. This study revealed high acceptability and usability of the COVID-19 antigen self-test among populations under socioeconomic vulnerability in Brazil. However, misinterpretation poses public

**Data availability statement:** Dataset provided as supplementary material.

**Funding:** We would like to thank the Brazilian Ministry of Health, UNITAID, and the World Health Organization (WHO) for supporting this project. UNITAID accelerates access to innovative health products and lays the foundations for their scale-up by countries and partners, and it is a hosted partnership of WHO. TST was financed in part by Conselho Nacional de Desenvolvimento Científico e Tecnológico (CNPq (#304417/2025-4 and #405558/2025-2), and Fundação de Amparo à Pesquisa do Estado do Rio de Janeiro (FAPERJ # E-26/201.270/2022). The funders of the study had no role in study design, data collection, data analysis, data interpretation, or writing of the report.

**Competing interests:** The authors have declared that no competing interests exist.

and individual health risks, underscoring the need of simplified, multimodal and accessible instructions.

## Background

Diagnostic testing remains pivotal for identifying and curbing the spread of severe acute respiratory syndrome coronavirus 2 (SARS-CoV-2). Rapid diagnostic self-tests emerged as a crucial tool during the pandemic, enabling large-scale testing and contributing to containment strategies [1]. Although the acute phase of the COVID-19 pandemic has subsided, the virus continues to circulate globally, underscoring the ongoing need for accessible and flexible testing approaches [2]. The lessons learned from COVID-19 highlight self-testing as a scalable public health strategy that can strengthen preparedness and response for both current and future infectious disease threats.

Self-sampling and self-testing have been widely incorporated into healthcare practices both for chronical [3] and infectious diseases, such as HIV [4,5]. Rapid diagnostic self-tests offer a user-friendly solution by providing results within minutes for individuals who test themselves in nonclinical settings. COVID-19 rapid diagnostic self-tests facilitate swift virus detection, allowing affected individuals to self-isolate promptly and reduce transmission risks [6]. Beyond their immediate use in pandemic response, these technologies represent a sustainable approach to decentralizing diagnostics and empowering individuals, particularly in settings where access to healthcare is limited or associated with stigma or logistical barriers [7].

The World Health Organization (WHO) has endorsed COVID-19 self-tests, [8] and multiple studies have demonstrated their accuracy and reliability [9–11]. However, their successful implementation depends on several factors, including accessibility, health literacy, and perceived efficacy by policymakers, clinical decision-makers and health care providers [1,7]. Among populations experiencing socioeconomic vulnerability, barriers to healthcare and information can compromise uptake and correct use [10].

Despite the approval of COVID-19 self-testing in Brazil in 2022, [6,12] knowledge gaps remain regarding its acceptability and usability, especially among populations experiencing socioeconomic vulnerability. Addressing these gaps is essential to optimize current self-testing policies and inform broader diagnostic strategies that promote equity and health literacy [6] Studies on acceptability and usability also help understand public perceptions of the value, utility, and pertinence of these tests. Evidence from the USA suggests that individuals with low socioeconomic status may be less motivated to use COVID-19 self-tests, [2] reinforcing the need to tailor self-testing strategies to social contexts and user needs.

COVID-19 rapid antigen self-tests hold lasting potential as part of a broader shift toward person-centered diagnostic models. In this sense, this study aimed to evaluate the acceptability and usability of a COVID-19 antigen self-test among populations living in socioeconomically vulnerable areas of Salvador and Rio de Janeiro, Brazil. This study contributes to understanding how self-testing can be effectively

implemented in resource-limited contexts. These insights may inform not only ongoing COVID-19 control efforts but also the design of equitable, community-based diagnostic strategies for future infectious disease outbreaks.

## Methods

### Study design

From January 2023 to May 2023, we conducted a cross-sectional study among individuals aged 18 or older recruited in three public primary healthcare services (PHC). One PHC was located in Cabula-Beirú (Salvador) and two PHC in Manguinhos (Rio de Janeiro). Healthcare professionals and persons self-declared illiterate were excluded. This study was part of the project *"Expansion of testing, quarantine, digital health, and telemonitoring strategies to tackle the COVID-19 pandemic in Brazil,"* which involved users from 19 public PHC located in neighborhoods under high socioeconomic vulnerability in Rio de Janeiro (Manguinhos) and Salvador (Cabula-Beirú). These neighborhoods are historically underserved territories characterized by high poverty, low education and income levels, limited access to basic services, and high exposure to urban violence. These factors reflect persistent socioeconomic vulnerability and social exclusion [13].

This study employed a mixed sampling approach. PHC were selected by convenience, as they were part of the broader project and served populations under high socioeconomic vulnerability. Within these PHC, participants were purposively recruited to ensure diversity in race (at least 50% Black and *Pardo* or mixed-race), gender (at least 50% women), and age (minimum proportion of 1/5 for each age group: 18–24 years, 25–34 years, 35–44 years, 45–59 years, and 60 + years). We established an intentional sample of 440 participants, including a 20% loss.

Potential participants were approached at the entrance of PHC by trained healthcare professionals after triage to determine whether they were seeking COVID-19–related services. Individuals who sought COVID-19 diagnostic testing were referred to testing services and were not invited to participate in the self-testing study.

### COVID-19 antigen self-test

We used the Panbio COVID-19 antigen self-test manufactured by Abbott®, which is a rapid diagnostic test designed to detect the presence of SARS-CoV-2 antigens in anterior nasal swab samples. There was an independent evaluation of the accuracy of this self-test, which showed an overall test sensitivity of 81% (95% confidence interval [CI] 75% to 86%) and specificity of over 99% compared to the reference laboratory's RT–PCR test [9]. In addition, the test was more sensitive (87%; 95% CI 80% to 91%) in symptomatic individuals than in asymptomatic individuals (71%; 95% CI 61% to 80%) [9].

### Procedures

A real-world testing scenario was simulated to evaluate the self-test device, which was presented in a ready-to-use package. Before the test, participants completed a sociodemographic questionnaire, which contained information about previous experiences with COVID-19 self-tests, administered by a study staff. Then, each participant was asked to consult the instructions for use in full before performing the self-test. Written and video instructions were available; the video was created for this study and was available in Portuguese language at: https://www.youtube.com/watch?v=TPED4BpFKn8. Participants were supervised by a study staff (healthcare professional) during all crucial steps, including test preparation, nasal sample collection and test execution. The study staff completed a structured evaluation form during self-test performance for each participant. As this study simulated a real-world testing scenario, the study staff was instructed not to interfere during self-testing but could provide written or video instructions again if requested by the participant. All procedures were performed on the same day and at the same place.

We evaluated the ability of the participants to read and interpret six standardized COVID-19 self-test results: three positive tests, one negative test, and two invalid tests. These standardized tests were coded by letters to determine the expected results (Fig 1). In addition, participants were asked to interpret their own self-test result, which was evaluated by

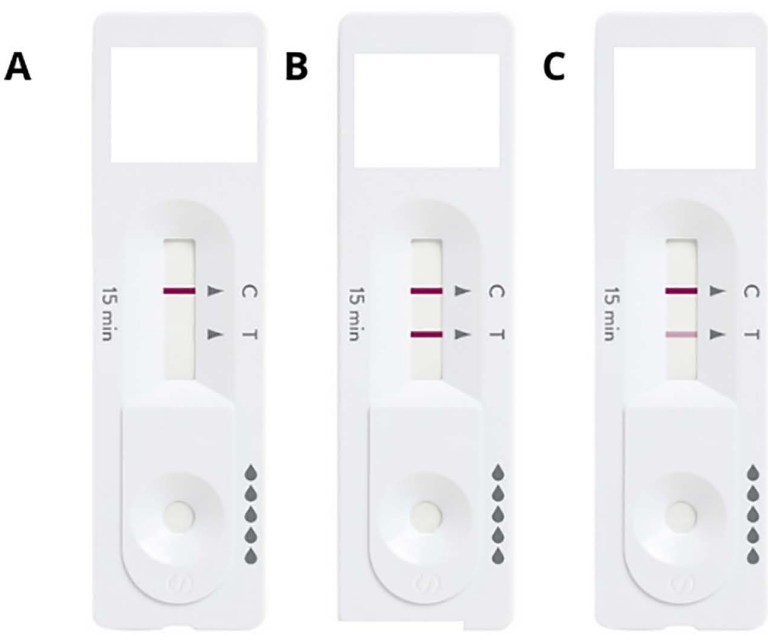

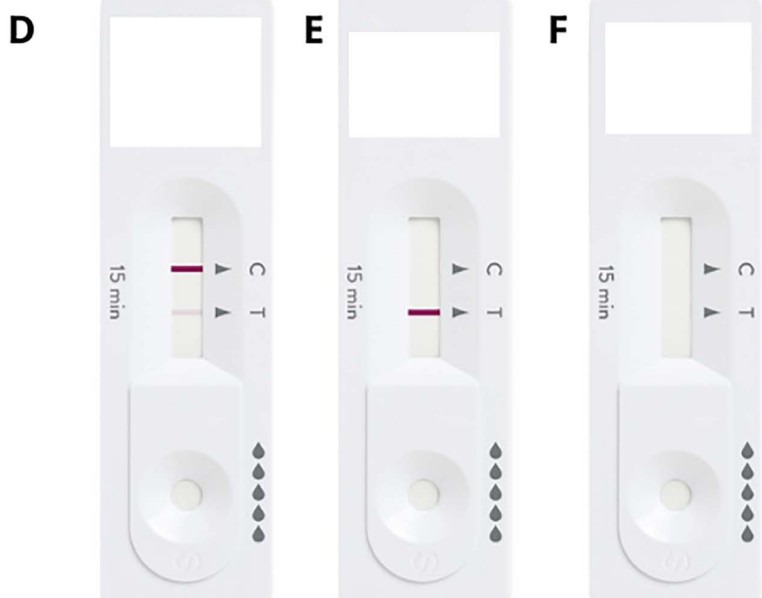

**Fig 1. Standardized COVID-19 self-test results.** Footnote: Tests were presented for interpretation by the participants separately during the study.

the study staff afterwards. Both interpretations were collected, and the participants received the correct test result at the end of the visit. In case of positive result, participants were referred to a health care professional in the same facility.

Lastly, participants were asked about acceptability of COVID-19 self-test as follows: "How was your experience using the COVID-19 self-test?", with possible answers ranging from "very bad" to "very good" (5-point Likert scale); "Would you use the COVID self-test again if necessary?" (Yes/No); "Would you recommend the COVID-19 self-test to someone else?" (Yes/No); "On a scale of 0 to 5, where 0 means "very difficult" and 5 means "very easy", how would you rate your experience using the self-test?"; and "Do you think the information received about the self-test was sufficient?" (Yes/No).

### Statistical analysis

We first described the sociodemographic characteristics of study participants overall and by location (Rio de Janeiro and Salvador), with differences in composition between cities analyzed using Pearson chi-square test. We then described COVID-19 self-test usability performance and test results overall and by location, providing proportions, 95% CI, and differences in composition between cities analyzed using Pearson chi-square test.

To evaluate the comprehension of instructions and proper execution of the COVID-19 self-test, we used Poisson regression models with robust variance to assess factors associated with obtaining a valid COVID-19 self-test result. A valid result was defined as the presence of a control line, indicating that participants performed the test correctly. Results were considered statistically significant at a 95% CI. The initial set of independent variables considered for inclusion in the adjusted model included city, gender, race, age group, education level, part of cash transfer program (Bolsa Família), income level, and previous use of COVID-19 self-test. Stepwise backward selection was used to identify the final set of variables for the adjusted model. To account for potential multicollinearity among the predictors, the variance inflation factor (VIF) was calculated, and no variables exceeded the conventional threshold (VIF > 5), indicating minimal multicollinearity. Adjusted prevalence ratios (aPR) and their 95% CIs were estimated from the final Poisson model with robust standard errors. The variable "watching the instructional video" was not included in the model due to low variability, as only a small subset of participants did not watch the video (n = 6).

We evaluated the accuracy in interpretation of COVID-19 self-test results overall and per sociodemographic characteristic. We first provided the proportion of correct answers for six standardized tests (one negative, three positive and two invalid). Then, we provided the proportion of concordance between the participants' test results and the correct result provided by the study staff. We used Cohen's kappa statistic, which accounts for random agreement and the possibility of guessing by some participants due to uncertainty: <0, poor agreement; 0.00–0.20, slight agreement; 0.21–0.40, fair agreement; 0.41–0.60, moderate agreement; 0.61–0.80, substantial agreement; and >0.8, almost perfect agreement [14].

Lastly, we described in terms of absolute number and proportion the acceptability of COVID-19 self-test overall and per location. We used Pearson chi-square test to verify differences per location. All analyses were performed using R project (version 4.4.1).

### Ethics approval and consent to participate

This study was part of the TQT project, [15] and the protocol was approved by the research ethics committees of the WHO (protocol identification numbers CERC.0128A and CERC.0128B) and the local Brazilian Institutional Review Boards at each site (protocol identification in Salvador, ISC/UFBA number 53844121.1001.5030; and Rio de Janeiro, INI/Fiocruz number 53844121.4.3001.5240, ENSP/Fiocruz number 53844121.4.3001.5240, and SMS/RJ number 53844121.4.3002.5279). All participants signed an informed consent form before starting the study.

### Results

After 444 individuals were initially included, seven opted not to continue after responding to the sociodemographic questionnaire. In total, 437 individuals were included in this study, 229 (52.4%) in Rio de Janeiro and 208 (47.6%) in Salvador.

The majority were women (65.7%; including one trans woman), Black or *Pardo* (81.5%), aged 35 years or older (65.7%), had a total household income of two Brazilian minimum wages or less (70.0%), and had completed secondary or incomplete superior education level (46.2%). The characteristics of study population differed significantly between the locations concerning gender, with more men in Rio de Janeiro; race, with more Black persons in Salvador; income level, higher in Rio de Janeiro; and education, higher in Salvador (Table 1).

Table 2 shows COVID-19 self-test usability performance and test results. Most of the procedures were performed correctly by at least 70% of the individuals. Participants from Salvador consistently showed better results than those from Rio de Janeiro in the correct execution of procedures. Some crucial procedures proved to be more challenging, such as the correct removal of the swab from the package (performed correctly by 46.3% in Rio de Janeiro), rotating the swab at least 5 times in each nostril (61.1% in Rio de Janeiro) and waiting the minimum time (15 minutes) to read the result (27.5% in Rio de Janeiro, 15.9% in Salvador, and 22.1% overall). Although the procedures were not always followed perfectly, most

**Table 1. Sociodemographic characteristics of study participants overall and by location (N = 437).**

| Socioeconomic characteristics | Total N (%) | Rio de Janeiro n (%) | Salvador n (%) | p-value |
|---|---|---|---|---|
| **Total** | 437 (100) | 229 (52.4) | 208 (47.6) | |
| **Gender** | | | | <0.0001 |
| Woman[1] | 287 (65.7) | 126 (55.0) | 161 (77.4) | |
| Man | 150 (34.3) | 103 (45.0) | 47 (22.6) | |
| **Race** | | | | 0.006 |
| White | 73 (16.7) | 51 (22.3) | 22 (10.6) | |
| Black | 160 (36.6) | 74 (32.3) | 86 (41.3) | |
| *Pardo* (mixed) | 196 (44.9) | 100 (43.7) | 96 (46.2) | |
| Indigenous or Asian | 8 (1.8) | 4 (1.7) | 4 (1.9) | |
| **Age group (years)** | | | | 0.962 |
| 18-24 | 74 (16.9) | 38 (16.6) | 36 (17.3) | |
| 25-34 | 76 (17.4) | 39 (17.0) | 37 (17.8) | |
| 35-44 | 69 (15.8) | 34 (14.8) | 35 (16.8) | |
| 44-59 | 92 (21.1) | 50 (21.8) | 42 (20.2) | |
| 60+ | 126 (28.8) | 68 (29.7) | 58 (27.9) | |
| **Cash transfer program (*Bolsa Família*)** | | | | 0.047 |
| Yes | 54 (12.4) | 21 (9.2) | 33 (15.9) | |
| **Income level[2]** | | | | <0.0001 |
| < 1 minimum wage (<US$235) | 107 (24.5) | 37 (16.2) | 70 (33.6) | |
| 1-2 minimum wages (US$235–471) | 199 (45.5) | 107 (46.7) | 92 (44.2) | |
| >2–5 minimum wages (>US$471–1178) | 104 (23.8) | 66 (28.8) | 38 (18.3) | |
| >5–10 minimum wages (>US$1178–2356) | 24 (5.5) | 18 (7.9) | 6 (2.9) | |
| > 10 minimum wages (> US$2356) | 3 (0.7) | 1 (0.4) | 2 (1.0) | |
| **Educational Level** | | | | 0.022 |
| Never studied or incomplete primary | 116 (26.6) | 59 (25.7) | 57 (27.4) | |
| Primary or incomplete secondary | 77 (17.6) | 51 (22.3) | 26 (12.5) | |
| Secondary or incomplete superior | 202 (46.2) | 103 (45.0) | 99 (47.6) | |
| Superior or higher | 42 (9.6) | 16 (7.0) | 26 (12.5) | |

[1]Including one trans woman.

[2]Monthly household income; values refer to the Brazilian minimum wage in 2022 and consider the exchange rate of 23 December 2022.

**Table 2. COVID-19 self-test usability performance and test results overall and by location: Salvador and Rio de Janeiro (N = 437).**

| | Total | | Rio de Janeiro | | Salvador | | |
|---|---|---|---|---|---|---|---|
| | N | % (95% CI) | N | % (95% CI) | N | % (95% CI) | p-value |
| **Use of the Covid-19 self-test** | | | | | | | |
| Using the leaflet | 19 | 4.3 (2.7 to 6.8) | 16 | 7.1 (4.2 to 11.3) | 3 | 1.4 (0.4 to 4.5) | <0.001 |
| Watching instructional video | 433 | 99.1 (97.5 to 99.7) | 225 | 98.3 (95.3 to 99.4) | 208 | 100.0 (97.7 to 100.0) | 0.16 |
| Restraining from touching the inside of the material | 398 | 91.1 (87.9 to 93.5) | 210 | 91.7 (87.2 to 94.8) | 188 | 90.4 (85.3 to 93.9) | 0.75 |
| Handling the testing device from the side | 386 | 88.3 (84.9 to 91.1) | 204 | 89.1 (84.1 to 92.7) | 182 | 87.5 (82.0 to 91.5) | 0.71 |
| Removing the testing device correctly | 379 | 87.7 (83.1 to 89.7) | 205 | 89.5 (84.6 to 93.0) | 174 | 83.7 (77.8 to 88.3) | 0.13 |
| Ensuring that the tube with the swab has been placed in an upright position | 338 | 77.3 (73.1 to 81.1) | 186 | 81.2 (75.4 to 85.9) | 152 | 73.1 (66.4 to 78.9) | 0.07 |
| Placing the testing device correctly on the table | 415 | 95.1 (92.4 to 96.7) | 217 | 94.8 (91.1 to 97.1) | 198 | 95.2 (91.1 to 97.5) | 1.00 |
| Restraining from touching the test strip or the place where the drops are dripped | 370 | 84.7 (80.9 to 87.8) | 190 | 83.1 (77.3 to 87.5) | 180 | 86.5 (81.0 to 90.7) | 0.37 |
| Keeping the testing device in a horizontal position for the duration of the test | 414 | 94.7 (92.1 to 96.6) | 216 | 94.3 (90.3 to 96.8) | 198 | 95.2 (91.1 to 97.5) | 0.85 |
| Removing the cap from the dropper bottle by holding it directly over the hole in the testing device | 393 | 89.9 (86.6 to 92.5) | 198 | 86.5 (81.2 to 90.5) | 195 | 93.8 (89.3 to 96.5) | 0.02 |
| Placing the correct number of drops in the top hole without touching the card with the tip | 356 | 81.5 (77.4 to 84.9) | 174 | 76.1 (69.8 to 81.3) | 182 | 87.5 (82.0 to 91.5) | <0.001 |
| Removal of the swab from the package holding the stick by the tip | 283 | 64.8 (60.1 to 69.2) | 106 | 46.3 (39.7 to 53.0) | 177 | 85.1 (79.4 to 89.5) | <0.001 |
| Rotating the same swab at least 5 times in each nostril | 296 | 67.7 (63.1 to 72.1) | 140 | 61.1 (54.5 to 67.4) | 156 | 75.0 (68.4 to 80.6) | <0.001 |
| Positioning the tube in the holder and transferring the contents of the bottle with the liquid | 408 | 93.4 (90.5 to 95.4) | 206 | 90.1 (85.1 to 93.4) | 202 | 97.1 (93.5 to 98.8) | <0.001 |
| Inserting the swab into the tube, rotating it 5 or more times and breaking the swab | 323 | 73.9 (69.5 to 77.9) | 146 | 63.8 (57.1 to 69.9) | 177 | 85.1 (79.4 to 89.5) | <0.001 |
| Dripping 5 or more drops into the round part of the device | 363 | 83.1 (79.1 to 86.4) | 178 | 77.8 (71.7 to 82.8) | 185 | 88.9 (83.7 to 92.7) | 0.01 |
| Waiting 15 minutes to read the result | 96 | 22.1 (18.2 to 26.2) | 63 | 27.5 (21.9 to 33.9) | 33 | 15.9 (11.3 to 21.7) | <0.001 |
| Waiting less than 20 minutes to read the result | 415 | 95.1 (92.4 to 96.7) | 210 | 91.7 (87.2 to 94.8) | 205 | 98.6 (95.5 to 99.6) | <0.001 |
| **Obtained a valid COVID-19 self-test result[1]** | | | | | | | |
| Yes | 385 | 88.1 (84.6 to 90.9) | 187 | 81.7 (75.9 to 86.3) | 198 | 95.2 (91.1 to 97.5) | <0.001 |
| **Test result** | | | | | | | |
| Positive | 9 | 2.4 (1.1 to 4.5) | 2 | 1.1 (0.2 to 4.2) | 7 | 3.5 (1.6 to 7.4) | NA |
| Negative | 376 | 86.0 (82.4 to 89.1) | 185 | 98.9 (95.8 to 99.8) | 191 | 96.5 (92.6 to 98.4) | NA |

NA: not applicable; [1] A valid result was defined as the presence of a control line in the self-test result, indicating that participants were able to perform the test correctly.

participants obtained a valid COVID-19 self-test result (81.7% in Rio de Janeiro, 95.2% in Salvador, and 88.1% overall). Among the valid results, the positivity rate was 2.4% overall, 1.1% in Rio de Janeiro and 3.5% in Salvador.

Regarding the socioeconomic factors associated with obtaining a valid COVID-19 self-test result, no multicollinearity was detected among the independent variables. Participants aged 35 or older had a significantly lower probability of obtaining a valid result compared to those aged 18–24 years: 35–44 years (aPR 0.89; 95% CI 0.82–0.98), 45–59 years (aPR 0.89; 95% CI 0.81–0.97), and 60 years or older (aPR 0.89; 95% CI 0.82–0.96) (Table 3). Participants with lower education level also had significantly reduced probability of obtaining a valid result compared to those with higher education. Participants who had never attended school or who did not complete primary education had an aPR of 0.83 (95% CI 0.75–0.92), and those who completed primary or incomplete secondary education had an aPR of 0.84 (95% CI 0.76–0.95). Women were significantly more likely to have valid results than men (aPR 1.10; 95% CI 1.02–1.19).

Overall, participants with older age and lower education levels consistently showed greater difficulty interpreting standardized test results, particularly in cases involving weak positive markers or invalid tests with misleading markings (Table 4). Participants generally performed better when interpreting negative results, with approximately 78% correctly identifying a standardized negative test card (Table 4; Fig 1A). Participants had more difficulty interpreting standardized tests with positive results, particularly when the positive marker line was lighter. For Positive 1 (two strong lines – Fig 1B),

**Table 3. Factors associated with obtaining a valid COVID-19 self-test result, with prevalence ratios (PR) and adjusted prevalence ratios (aPR) (N = 437).**

| | P value | PR (CI 95%) | P value | aPR (CI 95%) |
|---|---|---|---|---|
| **City (reference = Rio de Janeiro)** | | | | |
| **Salvador** | 0.00 | **1.16 (1.08-1.24)** | | |
| **Gender (reference = Men)** | | | | |
| Woman | **0.02** | **1.10 (1.02-1.19)** | 0.02 | **1.10 (1.02-1.19)** |
| **Race (reference = White)** | | | | |
| Black | 0.24 | 1.07 (0.96-1.19) | | |
| *Parda* | 0.49 | 1.04 (0.93-1.16) | | |
| Indigenous and Asian | 0.00 | 1.18 (1.07-1.30) | | |
| **Age group (reference = 18–24 years)** | | | | |
| 25-34 years | 0.06 | 0.93 (0.87-1.00) | **0.02** | **0.92 (0.85-0.99)** |
| 35-44 years | **0.02** | **0.90 (0.82-0.98)** | 0.02 | **0.89 (0.82-0.98)** |
| 45-59 years | **0.00** | **0.87 (0.80-0.95)** | 0.00 | **0.89 (0.81-0.97)** |
| 60 + Years | **0.00** | **0.84 (0.78-0.92)** | 0.00 | **0.89 (0.82-0.96)** |
| **Educational level (reference = Superior and higher)** | | | | |
| Never studied or incomplete primary | **0.00** | **0.83 (0.75-0.92)** | 0.00 | **0.83 (0.75-0.92)** |
| Primary or incomplete secondary | **0.00** | **0.85 (0.76-0.95)** | 0.00 | **0.84 (0.75-0.95)** |
| Secondary or incomplete superior | 0.16 | 0.96 (0.90-1.02) | 0.06 | 0.94 (0.88-1.00) |
| **Cash transfer program (Bolsa Família) (reference = No)** | | | | |
| Yes | 0.43 | 0.95 (0.85-1.07) | | |
| **Income level (reference > 5 minimum wages (> U$1178)[1]** | | | | |
| < 1 minimum wage (<US$235) | 0.42 | 0.95 (0.83-1.08) | | |
| <1-2 minimum wages (>US$235–471) | 0.50 | 0.96 (0.85-1.08) | | |
| <2-5 minimum wages (>US$471–1178) | 0.48 | 0.96 (0.84-1.09) | | |
| **previous use of COVID-19 self-test (reference = Yes)** | | | | |
| No | 0.99 | 1.00 (0.85-1.18 | | |

[1]Monthly household income; values refer to the Brazilian minimum wage in 2022 and consider the exchange rate of 23 December 2022.

PLOS Global Public Health

**Table 4. Accuracy in interpretation of COVID-19 self-test results overall and per sociodemographic characteristic (N = 437).**

| | % participants with correct answer to standardized tests | | | | | | Participant's result | |
|---|---|---|---|---|---|---|---|---|
| | **Negative** | **Positive 1**[1] | **Positive 2**[2] | **Positive 3**[3] | **Invalid 1**[4] | **Invalid 2**[5] | **% concordance** | **Cohen Kappa** |
| **Total** | 78.0 (73.8 to 81.8) | 63.6 (58.9 to 68.1) | 32.9 (28.6 to 37.6) | 25.2 (21.2 to 29.6) | 32.7 (28.4 to 37.4) | 77.1 (72.8 to 80.9) | 89.6 (86.1 to 92.1) | 0.56 (0.37 to 0.76) |
| **Gender** | | | | | | | | |
| Woman | 79.4 (74.2 to 83.9) | 65.5 (59.7 to 70.9) | 31.7 (26.4 to 37.5) | 24.1 (19.0 to 29.1) | 31.7 (26.4 to 37.5) | 77.4 (72.0 to 82.0) | 91.0 (86.9 to 93.9) | 0.45 (-0.15 to 0.75) |
| Men | 75.3 (67.5 to 81.8) | 60.0 (51.7 to 67.8) | 35.3 (27.8 to 43.6) | 27.8 (21.1 to 36.0) | 35.2 (27.2 to 42.9) | 76.7 (68.9 to 83.0) | 87.0 (79.9 to 91.5) | 0.73 (0.40 to 1.0) |
| **Age group (years)** | | | | | | | | |
| 18-24 | 91.9 (82.6 to 96.7) | 77.0 (65.5 to 85.7) | 41.9 (30.7 to 53.9) | 33.8 (23.5 to 45.8) | 32.4 (22.3 to 44.4) | 94.6 (86.0 to 98.3) | 95.9 (87.8 to 98.9) | 0.39 (0.15 to 0.92) |
| 25-34 | 89.5 (79.8 to 95.0) | 78.9 (67.8 to 87.1) | 42.1 (31.0 to 54.1) | 23.7 (15.1 to 35.1) | 42.1 (31.0 to 54.1) | 89.5 (79.8 to 95.0) | 96.1 (88.1 to 99.1) | 0.68 (0.38 to 0.98) |
| 35-44 | 85.5 (74.5 to 92.5) | 75.4 (63.3 to 84.6) | 36.2 (25.2 to 48.8) | 27.5 (17.8 to 39.8) | 36.2 (25.2 to 48.8) | 89.9 (79.6 to 95.5) | 95.7 (87.1 to 99.0) | 0.78 (0.55 to 1.0) |
| 44-59 | 72.8 (62.4 to 81.3) | 62.1 (51.2 to 71.7) | 28.3 (19.6 to 38.8) | 22.8 (15.1 to 33.1) | 35.9 (26.3 to 46.6) | 78.3 (68.2 to 85.9) | 89.1 (80.5 to 94.4) | 0.65 (0.45 to 0.85) |
| 60+ | 62.7 (53.6 to 71.0) | 41.3 (32.7 to 50.4) | 23.8 (16.9 to 32.4) | 21.4 (14.8 to 29.8) | 23.0 (16.2 to 31.5) | 51.6 (42.6 to 60.5) | 81.0 (72.8 to 87.2) | 0.46 (0.28 to 0.63) |
| **Educational level** | | | | | | | | |
| Never studied or incomplete primary | 60.3 (50.8 to 69.2) | 44.8 (35.7 to 54.3) | 27.6 (19.9 to 36.8) | 25.9 (18.4 to 35.0) | 22.4 (15.4 to 31.3) | 61.2 (51.7 to 70.1) | 82.8 (74.4 to 88.9) | 0.43 (0.39 to 1.01) |
| Complete primary or incomplete secondary | 77.9 (66.8 to 86.3) | 58.4 (46.6 to 69.4) | 40.3 (29.4 to 52.1) | 32.5 (22.5 to 44.2) | 23.4 (14.8 to 34.7) | 70.1 (58.5 to 79.6) | 83.8 (72.5 to 90.4) | 0.45 (-0.08 to 1.26) |
| Complete secondary or incomplete | 84.2 (78.2 to 88.8) | 71.8 (65.0 to 77.8) | 31.7 (25.4 to 38.7) | 20.3 (15.1 to 26.6) | 40.1 (33.3 to 47.2) | 85.1 (79.3 to 89.6) | 95.5 (91.4 to 97.8) | 0.46 (0.18 to 1.27) |
| Superior or higher | 97.6 (85.9 to 99.8) | 85.7 (70.8 to 94.1) | 40.5 (26.0 to 56.7) | 33.3 (20.0 to 49.6) | 42.9 (28.1 to 58.9) | 95.2 (82.6 to 99.2) | 97.6 (85.9 to 99.8) | 0.46 (0.40 to 1.56) |

[1]Positive with two strong lines.

[2]Positive with a strong line and an intermediate line.

[3]Positive with one strong and one weak line.

[4]Inconclusive with a mark on the positive line.

[5]Inconclusive without marking.

the overall correct interpretation rate was 63.6%, but accuracy was lower among participants aged 60+ years (41.3%) and those with lower education levels, with 44.8% correct for those who never studies or with incomplete primary education and 58.4% for those who had completed primary or incomplete secondary education. Positive test 2 (one strong and one intermediate line – Fig 1C) was interpreted correctly by only 32.9% of participants, with the lowest accuracy observed among those aged 44–59 years (28.3%), those aged 60+ years (23.8%), and individuals who never studied or incomplete primary education (27.6%). Positive test 3 (one strong and one weak line – Fig 1D) was even more challenging, interpreted correctly by only 25.2% of participants, and the worst performance recorded among participants aged 60+ years

(21.4%) (Table 4). When assessing the ability to interpret invalid test results, participants had the most difficulty with Invalid Test 1 (inconclusive with a mark on the positive line – Fig 1E), with only 32.7% providing correct interpretations. Accuracy was particularly low among participants aged 60+ years (23.0%) and those with lower education, 22.4% among those who never studies or incomplete primary education and 23.4% among those with complete primary or incomplete secondary education. In contrast, Invalid Test 2 (inconclusive without any marking – Fig 1F) was interpreted correctly by 77.1% of participants. Even among older adults and those with the lowest education level, over half correctly identified this result (51.6% and 61.2%, respectively).

Accuracy in interpretation of test results was 89.6% overall and higher than 80% in all sociodemographic groups (Table 4). Results of Cohen's kappa tests were generally moderate, being 0.56 (95% CI 0.37 to 0.76) overall. Participants aged 25–34 years and 35–44 years had substantial agreement (0.68, 95% CI 0.38 to 0.98 and 0.78, 95% CI 0.55 to 1.0, respectively).

Regarding the acceptability of COVID-19 self-test, most participants reported having a good (65.7%) or very good (19.1%) experience (Table 5). Individuals from Salvador reported slightly better experiences than those from Rio de Janeiro. People over 35 years of age and those with lower education levels were slightly more likely to rate their experience as negative. Most participants (over 90%) would use the COVID-19 self-test again and recommended it to others, and this result was consistent across all sociodemographic characteristics and the two locations. Another consistent factor was that more than 90% of participants reported that the information they received was sufficient to perform the test. In addition, according to the observing study staff, approximately 40% of the respondents across all sociodemographic groups and at both locations had no difficulties in using the self-test. The most observed challenges were setting up the test (17.6%), collecting the nasal swab samples (16.4%), and interpreting the results (18.1%). Men consistently showed greater difficulty than women in all testing steps. Greater-than-average difficulty with test setup was noted among individuals aged 44–59 years and 60+ years (22% and 27.4%, respectively). Participants aged 18–24 years had more difficulties collecting nasal swab samples (20.3%) compared to older participants.

## Discussion

We evaluated the acceptability and usability of a COVID-19 antigen self-test in populations living in neighborhoods under socioeconomic vulnerability in Salvador and Rio de Janeiro, Brazil. Participants faced challenges in correctly executing certain test procedures, such as removing the swab from the package and interpreting faint positive test lines. Valid results were obtained by most participants, with Salvador showing higher success rates than the Rio de Janeiro site. Valid results differed according to age, education, and gender, with those over 35 years and lower education levels correlating with lower success rates. Despite procedural difficulties, most participants were willing to use again and recommend the COVID-19 self-test, with reported experiences being generally positive. Men and older individuals reported more difficulties, particularly in test setup and interpretation. Overall, socioeconomic factors such as education level significantly impacted the usability and interpretation of the COVID-19 self-test result.

The usability of COVID-19 nasal self-tests seems to be slightly superior to that of blood samples from a finger prick [10,16,17]. In our study, despite some problems with the handling of the materials, almost all participants managed to complete the test, in contrast to studies using fingerstick blood tests [16,18]. Our study also revealed that 88.1% of participants obtained a valid result, similar to studies from Malawi, [10] the United Kingdom, [16,18] and France [17]. This suggests a high level of reliability and accuracy of the COVID-19 self-test across diverse geographic and demographic contexts, indicating that these tests can be effectively used worldwide. The comparable success rates in different countries also highlight the potential for standardizing self-testing protocols and guidelines globally, which could simplify public health strategies and improve consistency in monitoring and controlling the spread of the virus.

The likelihood of obtaining a valid test in our study was associated with sociodemographic factors. This aligns with a study conducted in Malawi using the same COVID-19 self-test, which identified a linear trend showing higher odds of

**Table 5. Acceptability of COVID-19 self-test overall and per location (N = 431).**

| | Rio de Janeiro | | Salvador | | Overall | | |
|---|---|---|---|---|---|---|---|
| | N | % (CI 95%) | N | % (CI 95%) | N | % (CI 95%) | p-value |
| **Self-test experience rates** | | | | | | | 0.001 |
| Very good | 52 | 23.2 (17.9 to 29.3) | 30 | 14.5 (10.1 to 20.2) | 82 | 19.1 (15.5 to 23.1) | |
| Good | 129 | 57.3 (50.6 to 63.8) | 153 | 73.9 (67.3 to 79.6) | 282 | 65.3 (60.6 to 69.7) | |
| Fair | 32 | 14.2 (10.1 to 19.6) | 23 | 11.1 (7.3 to 16.4) | 55 | 12.7 (9.8 to 16.3) | |
| Poor | 7 | 3.1 (1.4 to 6.6) | – | – | 7 | 1.6 (0.7 to 3.5) | |
| Very bad | 5 | 2.2 (0.8 to 5.4) | – | – | 5 | 1.2 (0.4 to 2.8) | |
| **Would you use the Covid-19 self-test again** | 216 | 96.5 (92.3 to 98.0) | 205 | 99.0 (96.2 to 99.8) | 421 | 97.5 (95.4 to 98.7) | 0.13 |
| **Would you recommend the Covid-19 self-test** | 218 | 96.9 (93.4 to 98.6) | 199 | 96.1 (92.3 to 98.2) | 417 | 96.5 (94.2 to 98.0) | 0.13 |
| **Sufficient information about the test** | 221 | 98.2 (95.2 to 99.4) | 201 | 97.1 (93.5 to 98.8) | 422 | 97.7 (95.6 to 98.8) | 0.70 |
| **Difficulties in using the self-test** | | | | | | | |
| None | 95 | 42.2 (35.7 to 49.1) | 98 | 47.3 (40.4 to 54.4) | 193 | 44.7 (39.9 to 49.5) | 0.28 |
| How to set up the test | 45 | 20.0 (15.1 to 25.9) | 31 | 15.1 (10.5 to 20.7) | 76 | 17.6 (14.2 to 21.6) | 0.24 |
| How to collect the nasal swab samples | 43 | 19.1 (14.3 to 25.1) | 28 | 13.5 (9.3 to 19.1) | 71 | 16.4 (13.1 to 20.3) | 0.17 |
| How to interpret the result | 28 | 12.4 (8.6 to 17.6) | 50 | 24.2 (18.6 to 30.7) | 78 | 18.1 (14.6 to 22.1) | 0.002 |
| Waiting time for the result | 6 | 2.7 (1.1 to 6.1) | 11 | 5.3 (2.8 to 9.6) | 17 | 3.9 (2.4 to 6.4) | 0.23 |
| The explanatory video | 11 | 4.9 (2.6 to 8.8) | 2 | 0.9 (0.1 to 3.8) | 13 | 3.0 (1.7 to 5.2) | 0.037 |

obtaining valid test results with increasing levels of education [10]. In a study using COVID-19 self-test requiring blood sample from a finger prick among 5,328 participants from England, people from lower socioeconomic levels were more likely to obtain an invalid test result, and women were more likely to obtain a valid result, similar to our findings [18]. The fact that sociodemographic factors are associated with obtaining a valid result for both nasopharyngeal and fingertip blood tests suggests that educational level plays a crucial role in the accuracy of self-administered tests. Higher education levels likely correlate with better understanding and execution of test instructions, leading to valid results. The consistent finding across different studies that women are more likely to obtain valid results indicates potential gender differences in following testing procedures or understanding test instructions [18,19]. The differences in valid test results between Salvador and Rio de Janeiro in our study might be explained by the higher inclusion of men and individuals with more education in Salvador's participant group. This highlights the need for targeted educational campaigns and support to ensure accurate test results across all demographic groups.

Compared to studies conducted in populations with higher income or higher education, our findings suggest that barriers to COVID-19 self-testing in deprived settings are more strongly linked to health literacy, familiarity with diagnostic technologies, and access to clear visual or digital instructions. In contrast to participants in studies from the United Kingdom

and France [16–18], our participants more frequently struggled with procedural steps such as handling materials and interpreting faint positive lines, indicating that small differences in design or guidance materials may disproportionately affect those with lower education or limited experience with self-administered tests. However, in comparison with studies from Malawi and other low-resource contexts using the same test [10], our results demonstrate similar overall usability but a slightly greater influence of education and age, possibly reflecting the heterogeneity of urban deprivation in Brazil, where functional literacy and digital access vary widely even within neighborhoods. These contrasts emphasize that usability and acceptability cannot be assumed to generalize across socioeconomic settings, and that future self-testing strategies must account for contextual barriers such as limited health literacy, low trust in technology, and reduced access to digital or printed instructions. Such insights are essential to develop self-testing tools that remain reliable and inclusive when deployed in real-world, resource-constrained environments.

In our study, the interpretation of the test results was challenging. This issue, particularly the difficulty in interpreting positive results with faint lines, has also been reported in other studies evaluating the use of COVID-19 self-tests as mass screening tools in public health [20,21]. Importantly, this does not appear to be specific to the type of test used in this study, as similar problems have been observed with self-tests requiring a blood sample from a finger prick [16,18]. In contrast, it is worth noting that in another study using the Panbio COVID-19 antigen self-test, participants had no problems interpreting the tests correctly [10]. However, participants in that study were only presented with a clearly positive result (strong line, similar to Fig 1B) and an invalid result with no lines (similar to Fig 1F), both of which were reasonably well interpreted in our findings as well. A systematic review further supports our findings, identifying challenges in following instructions and interpreting faint positive test lines as common barriers to effective self-test use [21]. Another issue was that the instructional video did not clearly illustrate all possible appearances of positive results, which may have led to misinterpretation. The leaflet, which was used less, provided clearer examples, highlighting the need for future videos to provide more detailed instructions of all procedures.

Problems with the usability of the tests, both in handling and interpretation, may be related to their fast-track approval in the face of the enormous public health challenge posed by the SARS-CoV-2 pandemic [8,16]. Other types of self-tests widely used in health surveillance, such as HIV self-tests, went through several iterations before designs were appropriate for home use [4,22]. Therefore, the same levels of acceptability and usability for home-based self-testing for SARS-CoV-2 antigen using the rapid diagnosis self-tests cannot be assumed.

We found high acceptability of COVID-19 self-test, similar to international findings, [2,10,16–18] but greater than a survey conducted among 417 individuals in São Paulo, Brazil, in which only 40.2% were likely or very likely to use it [6]. Understanding acceptability rates helps better resource allocation. Areas with lower acceptability might require more intensive outreach and education efforts to ensure widespread adoption of self-testing.

Our findings highlight the importance of expanding access to COVID-19 self-testing within public health services and among vulnerable populations. By demonstrating high levels of acceptability and usability, even in low-resource settings, the results support the integration of self-testing as a tool to enhance early detection and reduce transmission. However, the observed difficulties in interpreting certain test results, particularly among older and individuals with lower education, underscore the need for targeted educational campaigns and clear instructional materials to ensure accurate result interpretation and effective use.

Our study is original because it focused on the acceptability and usability of the rapid diagnostic COVID-19 antigen self-test, which has not been evaluated focusing on populations under socioeconomic vulnerability of Brazil. This study can inform public health strategies by highlighting the unique needs and preferences of vulnerable populations. Understanding their acceptance and challenges with self-testing can lead to more effective health interventions and policies tailored to these groups, guiding resource allocation by identifying specific barriers to self-testing in vulnerable populations. However, this study had several limitations. The convenience sampling model meant that the cohort was not demographically (age, gender, race, geography) representative of the adult population of Rio de Janeiro or Salvador. Nevertheless, our

study population is expected to closely reflect the real population in these neighborhoods, where reliance on the public health system is high, especially in socioeconomically vulnerable areas. The study did not assess the feasibility of reporting cases to epidemiologic surveillance authorities, which could be a serious problem in Brazil [8,23] Since the data are reported by healthcare professionals rather than by the individuals performing the self-tests, a separate reporting channel would need to be established for self-tests. A limitation to the scalability of the self-testing strategy for COVID-19 based on our results is that the self-test relied on participants watching an instructional video, which may limit usability for individuals with low digital literacy or without access to compatible devices. Although our study was conducted in a supervised setting, in real life symptomatic individuals would likely use self-tests at home without assistance. If distributed through the public health system in Brazil, tests would be free and home-based, as seen in other programs such as HIV testing [24]. Because healthcare professionals in our study only observed without providing help, we expect the identified usability challenges to also occur in home settings.

## Conclusions

Our study shows high acceptability and overall good usability of a COVID-19 antigen self-test among populations under socioeconomic vulnerability in Brazil. However, misinterpretation poses public and individual health risks. Participants with older age and lower education were less likely to obtain valid or correctly interpreted results. This underscores the need for simplified, multimodal instructions (plain language, pictograms, on-pack graphics, brief videos) that explicitly depict faint positives and common invalid results. These instructions should be available offline and include QR/phone support. To minimize disparities at scale, programs should pair free distribution through PHC and community partners with options for assisted self-testing, age-friendly swabs/clearer read windows, and streamlined reporting with immediate linkage-to-care guidance. For future respiratory outbreaks, successful self-testing hinges on several key actions: performing usability tests with target groups before deployment, continuously tracking equity metrics (such as valid-result rates by age and education), and swiftly refining materials and test designs. These steps are essential to maximize effectiveness and minimize misinterpretation. The strong public willingness to reuse tests indicates self-testing is a viable public health strategy, provided these implementation safeguards are in place.

## Supporting information

**S1 Data. Dataset.**
(CSV)

## Acknowledgments

We thank the Municipal Health Secretariats, health professionals, community-based health agents, and local Salvador and Rio de Janeiro communities for supporting this project. We also thank all the participants for their time.

## Author contributions

**Conceptualization:** Debora Castanheira, Laio Magno, Ines Dourado, Valdilea G. Veloso, Thiago Silva Torres.

**Data curation:** Debora Castanheira.

**Formal analysis:** Debora Castanheira.

**Funding acquisition:** Ines Dourado, Valdilea G. Veloso.

**Investigation:** Fabiane Soares, Thiago Silva Torres.

**Methodology:** Debora Castanheira, Thiago Silva Torres.

**Project administration:** Debora Castanheira, Thais Aranha Rossi, Fabiane Soares, Daniele Novaes.

**Resources:** Laio Magno, Valdilea G. Veloso.

**Supervision:** Ines Dourado, Valdilea G. Veloso, Thiago Silva Torres.

**Validation:** Thiago Silva Torres.

**Visualization:** Fabiane Soares, Daniele Novaes.

**Writing – original draft:** Debora Castanheira, Thiago Silva Torres.

**Writing – review & editing:** Laio Magno, Thais Aranha Rossi, Suelen Seixas, Fabiane Soares, Daniele Novaes, Ines Dourado, Valdilea G. Veloso.

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
