## [Decision Letter · Decision Letter 0]

29 Aug 2025

PGPH-D-25-00874

Acceptability and Usability of COVID-19 self-test in populations under socioeconomic vulnerability in Brazil: a cross-sectional study

Dear Dr. Torres,

Thank you for submitting your manuscript to PLOS Global Public Health. After careful consideration, we feel that it has merit but does not fully meet PLOS Global Public Health’s publication criteria as it currently stands. Therefore, we invite you to submit a revised version of the manuscript that addresses the points raised during the review process.

Your manuscript has been assessed by 4 reviewers and they have provided comprehensive comments that can be found below. The main theme of the reviewer comments focuses on a need for better dissemination of the results in the discussion and what impact they may have.

Please review their comments carefully and make the appropriate revisions to your manuscript to address their concerns.

We look forward to receiving your revised manuscript.

Kind regards,

Emma Campbell, Ph.D

Staff Editor

Journal Requirements:

Additional Editor Comments (if provided):

Reviewer #1:

Reviewer #2:

Reviewer #3:

Reviewer #4:

Reviewers' comments:

Reviewer's Responses to Questions

**Comments to the Author**

1. Does this manuscript meet PLOS Global Public Health’s publication criteria ? Is the manuscript technically sound, and do the data support the conclusions? The manuscript must describe methodologically and ethically rigorous research with conclusions that are appropriately drawn based on the data presented.

Reviewer #1: Yes

Reviewer #2: Yes

Reviewer #3: Yes

Reviewer #4: Yes

2. Has the statistical analysis been performed appropriately and rigorously?

Reviewer #1: I don't know

Reviewer #2: I don't know

Reviewer #3: Yes

Reviewer #4: No

3. Have the authors made all data underlying the findings in their manuscript fully available (please refer to the Data Availability Statement at the start of the manuscript PDF file)?

Reviewer #1: Yes

Reviewer #2: No

Reviewer #3: No

Reviewer #4: No

4. Is the manuscript presented in an intelligible fashion and written in standard English?

Reviewer #1: Yes

Reviewer #2: Yes

Reviewer #3: Yes

Reviewer #4: Yes

5. Review Comments to the Author

Reviewer #1: In their study “Acceptability and Usability of COVID-19 self-test in populations under socioeconomic vulnerability in Brazil: a cross-sectional study” Castanheira et al. assess ease-of-use and approval of antigen-based self-tests for SARS-CoV-2 in deprived populations in Brazil. Overall, I find the study to be well conducted, with methods and results being described clearly. However, from reading the introduction and discussion, the novelty of the study does not become clear to me. I’ve provided suggestions for improving the interpretation of the study’s results below, together with a few minor comments to further enhance the manuscript.

Major comments

- Introduction: In the title, the authors clearly highlight the fact that the study was conducted in “populations under socioeconomic vulnerability” as one of the study’s key contributions to the scientific literature. To enhance readability and to emphasize novel aspects of the analysis, I would suggest to shorten the introduction, removing especially any aspects that do not directly relate to the topic of “antigen-based self-testing for SARS-CoV-2 in vulnerable populations”. Particularly, I would suggest to merge the second and third paragraph, and to shorten this merged paragraph to a maximum of three sentences. I would also suggest to merge the fourth and fifth paragraph, again condensing this merged paragraph to a maximum of 3-4 sentences.

- Discussion: As for the introduction, the discussion should have a clear focus on interpreting the study’s results in the context of using COVID-19 self-testing in deprived populations. I would ask the authors to state how their results differ compared to (1) studies that were conducted in settings with high income and a high level of education, (2) studies that we done in similar settings as the present study. I am aware that the authors already touch upon these aspects, e.g., in line 356 to 360, but would suggest to make “COVID-19 self-testing in deprived populations” a stronger focus of the discussion, and further explore *why* results might differ to other / similar settings. Furthermore, I would appreciate it if the authors could discuss the above-mentioned aspects without further increasing the word count of the discussion. Lastly, as a minor note, the authors speak of “challenges” in the discussion on several occasions. I find the term “challenges” vague, and would instead suggest to specifically describe the most relevant challenges encountered.

Minor comments

Title:

- I would suggest to specify that the SARS-CoV-2 tests assessed in the study are *antigen* tests.

Abstract:

- I would suggest to shorten the results section of the abstract along the lines of “… than in Rio de Janeiro (81.7%) (p<0.01). Participants showed difficulty interpreting test results, particularly the inconclusive with a positive mark (32.7% correct interpretations) and faint positive markers significantly reduced accuracy (below 40%). Accuracy in interpretation of test results was 89.6% overall, with moderate to substantial inter-rater agreement (Cohen’s kappa 0.56 overall, reaching 0.78 in some age groups). Over 90% of participants were willing to reuse and recommended the COVID-19 self-test. Higher age and lower educational status were associated with a reduced likelihood of obtaining a valid results and increased difficulties in test setup and interpretation.”

Main text:

- Line 76: The word “a” seems out of place and should be removed.

- Line 135: Please replace “showed” by “shown”

- Table 3: please replace “ref =” by “reference: “ at all instances. Also, there seems to be a closing-bracket missing at the end of the

- Table 4: what is meant by “faixa etaria”? Please use English language instead

- Line 309: add “as” for “… such as education …”

- Line 355: I assume it should say “antigen” instead of “antibodies”?

- Line 385 to 387: would delete as it does not stem from the study’s results

Conflict of interest:

- I would ask the authors to be more specific in outlining their conflict of interest. In which way / to what extent was the work supported by WHO, UNITAID and the Brazilian Ministry of Health?

Reviewer #2: In the manuscript, Acceptability and usability of COVID-19 self-test in populations under socioeconomic vulnerability in Brazil: a cross-sectional study, Castanheira et al. aimed to evaluate the acceptability of SARS-CoV-2 RDTs, by logistic regression, among people living in lower socioeconomic neighborhoods in two Brazilian cities in early 2023.

This work is useful for understanding the acceptability and usibility of RDT self-tests in a group of people with a certain socioeconomic status, a distinguishing characteristic which is currently missing in the literature. These results are informative and have the potential to inform future implementation and be broadly applicable to other self-testing scenarios; however, there are several revisions that I believe are necessary prior to publication. Major and minor revisions are outlined below.

Major revisions

1. Please check grammar and spelling throughout the manuscript.

2. How representative do you think your study population is relative to the general population of the neighborhoods you are studying?

3. Where there any factors that could have possibly created the challenges highlighted by the study population in Rio de Janeiro compared to Salvador (ie, packaging, instructions, etc.)?

4. Why is reporting cases an issue in Brazil? Please explain.

Minor revisions

1. I understand that the testing scenario was simulated, but is it accurate to assume that in real life participants who are likely sick and showing symptoms would be taking the test in the same environment that they are in the study? Would they be at a clinic with nurses there? Or would this likely happen at home alone? I would touch on this and how it might affect your results. This could also play into what you are saying about internet access to watch the YouTube video.

2. I think it is important to discuss where people access these tests and how that could play into usability and acceptability. Also, how are they accessible in terms of cost? Are people less likely to even bother getting a test due to their cost? Especially if they expect to not have a valid result?

3. Why do you expect your results differ from the study in Sao Paolo? Why is your study a necessary addition to the literature if there is already a study done in Brazil?

4. More discussion on the implications of these results is necessary. Can any of your results be generalized to future potential outbreak scenarios for other pathogens or maybe another pandemic? How can you ensure thoughtful implementation in your study group to minimize further disparities in this group? What would scale-up look like in a situation like this? Lay out what we need to do better.

Reviewer #3: General comments:

The authors of this study focus on the acceptability and usability of COVID-19 self-tests in populations who are socioeconomically vulnerable. It’s important to consider such populations when designing and implementing future public health interventions and the authors did well in discussing the studies implications.

Review manuscript text for grammatical errors.

Abstract:

The authors mention a simulated real-world setting in the abstract methods, the meaning of this was unclear until reading the main text. Participants were observed in the primary care setting by a healthcare provider. The authors could consider clarifying.

In the abstract results, the mention of age and education several times as factors associated with misinterpretation of results feels repetitive. The last sentence of the results could be moved to the conclusion, for example.

Background:

The perspective of the background section feels like it is referencing a prior stage of the COVID-19 pandemic. The dynamics of the pandemic have largely shifted, and large-scale public health interventions are no longer in place. While the use of self-tests is still important and valuable as SARS-CoV-2 remains prevalent, the paper may be better framed as setting us up for future success with self-testing more broadly, particularly among populations of socioeconomic vulnerability. I would recommend adding some commentary on how this work adds to the future of self-testing strategies.

Methods:

It may be important to mention that the Panbio COVID-19 antigen self-test uses an anterior nasal swab, as nasopharyngeal swabs were commonly used in early stages of the pandemic.

Methods are clearly described. Good use of logistic regression with VIF test of multicollinearity among the variables.

Results:

The results do not mention final variable selection from AIC in the logistic regression model. Do the variables shown in Table 3, with an aOR, correspond to the variables included in the multivariable model?

There was also no mention of the VIF results and multicollinearity between variables. Was any multicollinearity identified and were any variables removed from the regression model as a result?

Discussion:

The authors compared the study well with other literature. Conclusions drawn from the study are clear.

As was discussed, participants had difficulty interpreting self-test results with faint positive lines. Based on the instructional video linked to in the methods, they only show a self-test result with a strong positive line. As most participants relied on the video for instructions versus the leaflet, this may have been a factor in difficulty interpreting faint results. Does the leaflet show an example of a positive with a faint line? This could be important to mention in the discussion.

Reviewer #4: This paper is well written and well-scoped to add some valuable data from Brazil on COVID self-testing

Far more information is needed about recruitment procedures and implementation:

• Why were the cities of Salvador and Rio de Janeiro selected?

• What primary healthcare facilities were chosen and why? How many did study staff recruit at?

• How were participants actually selected? From a line list? How were they contacted? Who did the contacting and what was their background? How were the variables of race, gender, and age assessed?

• Convenience sampling and purposive sampling are two separate approaches, yet both are mentioned. This could use more explanation.

• The paper is presented as being about “socioeconomic vulnerability”, but it is not clear why participants on the study fall in this category, or the locations of recruitment represent this category of people.

• Who were the staff conducting the surveys? What was their background?

• Where did both recruitment and data collection happen? Was it the same time as the consent?

• For participants who tested positive on the self-test, were procedures any different? Were they still asked the acceptability questions and was that process any different? I would expect these participants could have been going through emotional stress at this point

Statistical analysis:

• Why were the variables for adjustment chosen? Were others considered and rejected?

• No mention of sample size consideration is present. Was there any sample size goal at the start of the study? why did the study stop recruitment at 444 participants?

Results

• Table 1 is not all in English

• The spacing of Table 2 makes it hard to read

• The outcome of interest has 88% prevalence (obtained a valid self-test). I’m not convinced that logistic regression is an appropriate model choice given this high outcome prevalence, as the odds ratios may be overestimated. Why is this justified? I suggest authors look into instead using log binomial or Poisson regression for prevalence ratios to see if that’s more appropriate since the outcome is common

• In table 5, I don’t think including the chi square test coefficient adds value

Discussion

• More discussion can be added about why such differences were seen between the study locations, it would also help to include more information in the methods section about the differences in context between locations

• More limitations to discuss are that these results are all in the context of supervision and further support from study staff – real world interpretation is limited without this help

6. PLOS authors have the option to publish the peer review history of their article (what does this mean? ). If published, this will include your full peer review and any attached files.

**Do you want your identity to be public for this peer review?** For information about this choice, including consent withdrawal, please see our Privacy Policy .

Reviewer #1: No

Reviewer #2: No

Reviewer #3: No

Reviewer #4: No

---

## [Decision Letter · Decision Letter 1]

27 Nov 2025

PGPH-D-25-00874R1

Acceptability and Usability of COVID-19 antigen self-test in populations under socioeconomic vulnerability in Brazil: a cross-sectional study

Dear Dr. Torres,

Thank you for submitting your manuscript to PLOS Global Public Health. After careful consideration, we feel that it has merit but does not fully meet PLOS Global Public Health’s publication criteria as it currently stands. Therefore, we invite you to submit a revised version of the manuscript that addresses the points raised during the review process.

We look forward to receiving your revised manuscript.

Kind regards,

Helen Howard

Staff Editor

Journal Requirements:

Additional Editor Comments (if provided):

Reviewers' comments:

Reviewer's Responses to Questions

**Comments to the Author**

1. If the authors have adequately addressed your comments raised in a previous round of review and you feel that this manuscript is now acceptable for publication, you may indicate that here to bypass the “Comments to the Author” section, enter your conflict of interest statement in the “Confidential to Editor” section, and submit your "Accept" recommendation.

Reviewer #1: All comments have been addressed

Reviewer #2: All comments have been addressed

Reviewer #3: All comments have been addressed

Reviewer #4: All comments have been addressed

2. Does this manuscript meet PLOS Global Public Health’s publication criteria ? Is the manuscript technically sound, and do the data support the conclusions? The manuscript must describe methodologically and ethically rigorous research with conclusions that are appropriately drawn based on the data presented.

Reviewer #1: Yes

Reviewer #2: Yes

Reviewer #3: Yes

Reviewer #4: Yes

3. Has the statistical analysis been performed appropriately and rigorously?

Reviewer #1: I don't know

Reviewer #2: I don't know

Reviewer #3: Yes

Reviewer #4: Yes

4. Have the authors made all data underlying the findings in their manuscript fully available (please refer to the Data Availability Statement at the start of the manuscript PDF file)?

Reviewer #1: Yes

Reviewer #2: Yes

Reviewer #3: Yes

Reviewer #4: Yes

5. Is the manuscript presented in an intelligible fashion and written in standard English?

Reviewer #1: Yes

Reviewer #2: Yes

Reviewer #3: Yes

Reviewer #4: Yes

6. Review Comments to the Author

Reviewer #1: I very much appreciate the author’s thoughtful consideration of the reviewers' comments to the manuscript! It is clear that great effort has gone into the revision, and I don’t have any further comments.

As a side note, I noticed two linguistic aspects while reading the manuscript that the authors may wish to consider if they find them helpful:

- Line 107: add a full stop after ‘(PHC)’ and start the new sentence with ‘… services (PHC). One PHC …’

- Line 127 / 128: what exactly is meant with individuals that ‘perchance tested positive’? Does this refer to individuals that were seeking COVID-19 related services and were initially excluded from the study, but were then included because of the positive result? Please clarify.

Reviewer #2: The authors have appropriately addressed all of my previous comments. Thank you.

Reviewer #3: Authors have sufficiently addressed all comments.

As authors have changed their regression method from logistic to Poisson, this needs to be updated in the abstract.

Reviewer #4: Great adjustments to the analysis and content in the paper! Thank you for addressing comments.

7. PLOS authors have the option to publish the peer review history of their article (what does this mean? ). If published, this will include your full peer review and any attached files.

**Do you want your identity to be public for this peer review?** For information about this choice, including consent withdrawal, please see our Privacy Policy .

Reviewer #1: No

Reviewer #2: No

Reviewer #3: No

Reviewer #4: No

 Figure Resubmissions:

---

## [Editor Report · Decision Letter 2]

4 Dec 2025

Acceptability and Usability of COVID-19 antigen self-test in populations under socioeconomic vulnerability in Brazil: a cross-sectional study

PGPH-D-25-00874R2

Dear Dr. Torres,

We are pleased to inform you that your manuscript 'Acceptability and Usability of COVID-19 antigen self-test in populations under socioeconomic vulnerability in Brazil: a cross-sectional study' has been provisionally accepted for publication in PLOS Global Public Health.

Best regards,

Julia Robinson

Executive Editor